# Vector Competence of *Aedes albopictus* for Yellow Fever Virus: Risk of Reemergence of Urban Yellow Fever in Brazil

**DOI:** 10.3390/v15041019

**Published:** 2023-04-21

**Authors:** Rossela Damasceno-Caldeira, Joaquim Pinto Nunes-Neto, Carine Fortes Aragão, Maria Nazaré Oliveira Freitas, Milene Silveira Ferreira, Paulo Henrique Gomes de Castro, Daniel Damous Dias, Pedro Arthur da Silva Araújo, Roberto Carlos Feitosa Brandão, Bruno Tardelli Diniz Nunes, Eliana Vieira Pinto da Silva, Lívia Carício Martins, Pedro Fernando da Costa Vasconcelos, Ana Cecília Ribeiro Cruz

**Affiliations:** 1Programa de Pós-Graduação em Biologia de Agentes Infecciosos e Parasitários, Universidade Federal do Pará, Belém 66075-110, PA, Brazil; 2Seção de Arbovirologia e Febres Hemorrágicas, Instituto Evandro Chagas, Secretaria de Vigilância e Saúde, Ministério da Saúde, Ananindeua 67030-000, PA, Brazil; 3Centro Nacional de Primatas, Instituto Evandro Chagas, Secretaria de Vigilância e Saúde, Ministério da Saúde, Ananindeua 67030-000, PA, Brazil; 4Programa de Pós’Graduação em Biologia Parasitária da Amazônia, Universidade do Estado do Pará, Belém 66087-662, PA, Brazil

**Keywords:** experimental infection, *Aedes albopictus*, yellow fever, reemergence, RT–qPCR, viral isolation

## Abstract

The risk of the emergence and reemergence of zoonoses is high in regions that are under the strong influence of anthropogenic actions, as they contribute to the risk of vector disease transmission. Yellow fever (YF) is among the main pathogenic arboviral diseases in the world, and the Culicidae *Aedes albopictus* has been proposed as having the potential to transmit the yellow fever virus (YFV). This mosquito inhabits both urban and wild environments, and under experimental conditions, it has been shown to be susceptible to infection by YFV. In this study, the vector competence of the mosquito *Ae. albopictus* for the YFV was investigated. Female *Ae. albopictus* were exposed to non-human primates (NHP) of the genus *Callithrix* infected with YFV via a needle inoculation. Subsequently, on the 14th and 21st days post-infection, the legs, heads, thorax/abdomen and saliva of the arthropods were collected and analyzed by viral isolation and molecular analysis techniques to verify the infection, dissemination and transmission. The presence of YFV was detected in the saliva samples through viral isolation and in the head, thorax/abdomen and legs both by viral isolation and by molecular detection. The susceptibility of *Ae. albopictus* to YFV confers a potential risk of reemergence of urban YF in Brazil.

## 1. Introduction

Yellow fever (YF) is among the main arboviral diseases in the world and is endemic in tropical regions of Africa and in several countries of South America, including Brazil, affecting the public health care in these countries [1,2,3]. It is an infectious, acute febrile disease caused by an arbovirus of the genus *Flavivirus* in the family *Flaviviridae*, i.e., it is the Yellow fever virus (YFV), which is transmitted by the bite of mosquitoes of the genus *Aedes*, especially *Aedes aegypti*, in urban areas, and the genera *Haemagogus* and *Sabethes* in wild areas in the Americas. It is estimated that there are 200,000 cases of YF and 30,000 deaths each year worldwide. Notably, 20% to 50% of those infected who develop severe disease die [4,5].

Environmental changes strongly affect the vector ecology and epidemiology of diseases. In tropical forest regions that are under the influence of anthropogenic actions, the risk of emergence and reemergence of zoonoses is high, considering that changes such as urbanization increase the risk of transmission of etiological agents by vectors. The dispersal of anthropophilic mosquitoes may be facilitated by changes in the ecosystem that affect the interactions between vectors and hosts and favor the action of vectors in the transmission of pathogens between wild and urban cycles [6,7,8].

Vector-borne diseases account for more than 17% of all infectious diseases worldwide, causing more than one million deaths annually. Viruses transmitted by arthropods belong to the group of arboviruses (*arthropod-borne* viruses), which are related to human pathogens, most of which are zoonotic. One of the vectors that has been widely investigated in the transmission of arboviruses is the *Aedes* (Stegomyia) *albopictus* [9], a mosquito native to the forests of Southeast Asia and found in Brazil for the first time in 1986, subsequently expanding to urban areas in recent years [6,10,11,12,13].

*Ae. albopictus* has been proposed as a potential transmission vector of YFV, in addition to other viruses, because it has a wide geographical distribution and ecological plasticity, a characteristic that allows procreation both in urban and wild environments, allowing this mosquito to become a bridge vector between urban environments and rural areas [6,8,14]. Although the vector competence of Brazilian populations of this Culicidae is still being investigated, in Asia, *Ae. albopictus* acts as a vector of the Dengue virus (DENV) and Japanese encephalitis virus (JEV) in rural and urban areas; in the Americas, it was found to be naturally infected with Venezuelan equine encephalitis virus (VEEV) and West Nile virus (WNV) and, under laboratory conditions, was found to be competent for at least 22 arboviruses, including DENV, Zika virus (ZIKV), Chikungunya virus (CHIKV) and YFV [2,15,16,17,18,19]. In studies conducted in Brazil, *Ae. albopictus* has also been found to be naturally infected with ZIKV [20,21] and DENV [21,22,23].

From 2000 to 2015, 339 human cases of sylvatic yellow fever (SYF) and 161 (47.4%) deaths were confirmed in Brazil. From December 2016 to May 2017, the country experienced the largest outbreak of SYF in recent decades, with 642 cases in non-human primates (NHPs) and 778 cases in humans, with 262 (34.59%) deaths reported in the states of the Midwest, Southeast, and North regions [1,24,25].

From 2018 to March 2019, 50 cases of YF were confirmed in the states of São Paulo and Paraná, 24% of which resulted in deaths. From 2019 to June 2022, there were 33 cases in the states of Pará, Tocantins, Acre and Santa Catarina, 12% of which resulted in deaths. This scenario has prioritized the organization of public health services, prevention activities, disease control and vector control and highlights the need to better understand the factors related to the dispersion of YFV [3,26].

The reemergence of YFV in endemic countries may be favored by a combination of several factors, such as the low demand and supply of the YF vaccine, the susceptibility of exposed human populations, the high density of vectors and primary hosts (i.e., NHPs), favorable climatic conditions, the emergence of a new genetic lineage and the circulation of infected people and NHPs [27,28,29].

Elucidating knowledge about the vector competence and the transmission of mosquito-borne infections improves the possibilities for preventing and responding to emerging threats from arboviruses as well as developing new warning systems and, consequently, vector control. The aim of this study was to evaluate, using NHPs of the genus *Callithrix* as an experimental model, the vector competence of *Ae. albopictus* and its potential role as a link between the sylvatic and urban cycles in a possible urban expansion of YFV in Brazil.

## 2. Materials and Methods

### 2.1. Mosquitoes

To conduct this study, populations of mosquitoes of the species *Ae. albopictus* were obtained from the municipality of Ananindeua, Pará, Brazil. The mosquitoes were kept until the fifth generation in an insectary of the Laboratory of Medical Entomology, Section of Arbovirology and Hemorrhagic Fevers, Evandro Chagas Institute (SAARB/IEC), under controlled temperature and humidity conditions (26 ± 1 °C; 80–90% RH) and a 12-h light–dark cycle. The larvae were fed fish chow, and adult mosquitoes were fed a 10% glucose solution.

For the experiment, the mosquitoes were deprived of food for 24 h before the blood meal. A total of 200 females were used, 5 to 7 days after emerging from the pupal stage, of which 50 were assigned to the control group (kept until the end of the experimental period).

### 2.2. NHPs

Ten NHPs of the genus *Callithrix* sp. (hybrid resulting from the crossing between the species *Callithrix penicillata* and *Callithrix jacchus*) bred and maintained at the National Primate Center (the CENP-body linked to the Secretary of Health and Environment Surveillance—SVS—of the Ministry of Health, which operates in the areas of conservation, reproduction and research with non-human primates) Ananindeua, Pará, Brazil, were used in this study. The choice for this hybrid animal model was due to the cases of YF in hybrid PNH’s of the genus *Callithrix* in epizootic outbreaks reported from 2017 to 2018 in different regions of Brazil. The animals were transported to the Animal Biosafety Level 3 (NBA3) Laboratory of the SAARB/IEC, strictly following the biosafety, containment and sanitary control recommendations [30,31,32] and were housed in 10 individual stainless steel cages with retractable bottoms (0.80 cm × 0.80 cm × 0.90 cm). During the entire experiment, the nutritional protocol recommended by the CENP was adopted for animal maintenance. All NHPs were previously tested by serological methods and were negative for flavivirus. One animal was maintained as a negative control until the end of the experimental period.

### 2.3. Virus

The isolate BeAN838691, belonging to the South American genotype I, obtained from the serum of a natural YFV infection in *Callithrix penicillata,* provided by SAARB/IEC, was used. The viral stock was prepared using *Ae. albopictus* (clone C6/36) and the first passage sample was inoculated into the NHPs [33]. Confirmation of the presence of YFV in the referred cells was performed using the indirect immunofluorescence assay [34], and the RNA copy number was obtained using RT–qPCR.

### 2.4. Infection Procedure

Three NHPs (NHP1, NHP2 and NFP3) were intramuscularly inoculated with a single dose of YFV suspension (1 × 105 PFU/mL) following the method described by Ferreira et al. [35]. On the 3rd day post-infection (dpi) of the NHPs, the animals were sedated with tiletamine-zolazepam (8 mg·kg^−1^) combined with chlorpromazine (0.5 mg·kg^−1^) [36]. Then, each NHP was exposed to 50 female *Ae. albopictus* (totaling 150 mosquitoes), which were retained in an adapted feeder that was placed in the abdominal region (hair previously removed by trichotomy) for a period of 30 min for blood feeding. After feeding, the engorged females were transferred to individual cages and maintained with 10% glucose up to 14 dpi (referring to the arthropod), when they were placed in contact with 3 healthy and sedated NHPs (NHP4, NHP5 and NHP6) for blood feeding, following the same conditions previously described for the first blood feeding. Subsequently, the females were distributed in individual cages, duly identified and maintained with 10% glucose up to 21 dpi. At this time, 3 healthy and sedated NHPs (NHP7, NHP8 and NHP9) were exposed to the female *Ae. albopictus* for blood feeding following the same protocol described above (Figure 1).

The 3 NHPs inoculated with YFV viral suspension and the 6 NHPs exposed to mosquitoes were euthanized on the 7th day after exposure to the virus using an overdose of intracardiac thiopental sodium (1 g) [37].

### 2.5. Collection of Mosquito Samples

At 14 and 21 dpi, the female *Ae. albopictus* had their saliva extracted and were segmented (head, thorax and abdomen). For the saliva extraction, the proboscis of each female was inserted into a 10 μL tip containing 5 μL of fetal bovine serum (FBS). The sample was pipetted into a 1.5 mL microtube containing 45 μL of Leibowitz L-15 medium and used for a cell culture [15,38]. The segmentation was performed individually using sterile plates, microscope slides, entomological forceps and a stereomicroscope with 14 and 21 dpi light. The abdomen and chest were examined to confirm infection, the head and legs were examined to confirm the spread, and the saliva was collected to verify the infection and possible transmission of YFV.

The samples were organized into batches of 10 specimens, placed in properly labeled tubes and ground in a Tissue Lyser II (Qiagen, Hilden, Germany) for 60 s at a frequency of 25 Hz in a 2 mL microtube (U-bottom) containing 1 mL of 1× D-PBS extender (Gibco, USA) with 2% penicillin and streptomycin, 1% fungizone, 5% FBS and a tungsten bead (3 mm). After grinding, the tubes were placed in a −80 °C freezer for 12 h.

### 2.6. Cell Culture, Inoculation and Indirect Immunofluorescence (IIF)

Following the protocol described by Igarashi [32], *Aedes albopictus* clone C6/36 cells were maintained at 28 °C using Leibowitz L-15 medium with L-glutamine (Gibco, USA) supplemented with 5% FBS (Gibco) (phosphate), tryptose phosphate (Himedia, Kennet Square, PA, USA) (2.95%), antibiotics (penicillin 10,000 U/L and streptomycin 10,000 µg/L) (Gibco) and nonessential amino acids (10 mL/L) (Baktron Microbiology, Rio de Janeiro, Brazil) [39]. Before the inoculation, the growth medium (5% FBS) was discarded from the tubes containing the cells, followed by the addition of 100 µL of the arthropod macerate solution. Sample adsorption was conducted for 60 min at 28 °C, with gentle shaking every 15 min.

After the sample was allowed to adsorb, 1.5 mL of L-15 maintenance medium was added to each tube containing the inoculated cells. The tubes were incubated again at 28 °C and observed daily for 7 days under an inverted optical microscope to visualize a possible cytopathogenic effect (CPE) [40,41]. On the 7th day after inoculation, IIF was performed following the protocol described by Gubler et al. [34].

For the IIF, the cells inoculated with the macerates were placed on slides and fixed in acetone PA (−20 °C) for 10 min. Subsequently, 10 μL of polyclonal antibodies against YFV provided by FIOCRUZ—Rio de Janeiro, Brazil, and diluted to 1:500 in a phosphate saline solution (PBS, pH 7.4) was added to each of the slides. Next, the slides were incubated for 30 min in an oven at 37 °C, followed by a PBS wash for 10 min. The slides were then allowed to dry at room temperature. Then, 10 μL of anti-mouse conjugated antibody (Cappel) diluted to 1:900 in PBS pH 7.4 was added to the slides, and the incubation and washing steps previously described were repeated. The slides were then mounted (coverslip and a small drop of buffered glycerin, pH 7.4) and observed under an epifluorescence microscope.

### 2.7. Quantification of the YFV Genome in Mosquitoes

#### 2.7.1. RNA Extraction

The total RNA was extracted from 140 μL of the macerate obtained during the collection stage using a QIAamp viral RNA^®^ kit (Qiagen, Hilden, Germany) following the manufacturer’s protocol. Bacteriophage MS2 RNA (Roche Diagnostics, Basel, Switzerland) was used as a noncompetitive exogenous internal control. For this purpose, 2 ng of MS2 RNA were added to each mosquito pool during extraction [42]. The extracted RNA was stored at −80 °C until use.

#### 2.7.2. Preparation of a YFV Standard Curve

To construct a YFV standard curve, a low-passage YFV isolate (strain BeAr843721, GenBank accession number MF370530) cultivated in Vero cells was used. Viral RNA was extracted from the cell culture supernatant using a QIAamp viral RNA^®^ kit (Qiagen) following the manufacturer’s instructions. Viral RNA was then quantified by RT-ddPCR using the One-step RT-ddPCR Advanced Kit for Probe (Bio-Rad, Hercules, CA, USA) and a set of primers and probe specific for YFV [43]. The RT-ddPCR mixtures contained 11 nmol of each primer, 4.4 nmol of probe and 5 μL of extracted viral RNA in a total volume of 22 μL. Next, the reaction mixtures were transferred to the wells of the DG8 droplet generator cartridge (Bio-Rad), to which 70 μL of droplet generator oil (Bio-Rad) were added. The reactions were emulsified in a QX-200 droplet generator (Bio-Rad). Then, 40 μL of emulsion/reaction were transferred to a 96-well reaction plate (ddPCR Plates 96-Well, Semi-Skirted, Bio-Rad). The plates were sealed in a PX1™ PCR Plate Sealer (Bio-Rad) and subjected to amplification in a C1000 Touch™ Thermal Cycler (Bio-Rad).

The cycling conditions were as follows: 95 °C for 10 min, followed by 40 cycles of denaturation at 95 °C for 30 s and elongation at 60 °C for 1 min. At the end of the amplification reaction, the PCR products were read using a Quanta Soft Droplet Reader (Bio-Rad). The absolute concentration of viral RNA, in copies/µL, was determined automatically using the ddPCR QuantaSoft Software V.1.7.4.0917 (Bio-Rad). After determining the absolute quantification of the viral RNA in copies of the YFV genome per microliter, the RNA was used as a standard to construct a standard curve for use in the quantification of the RNA extracted from the mosquito pools. The extracted viral RNA was serially diluted to 1:10, ranging from 3.3 × 10^4^ to 3.3 × 10^0^ copies/µL. The standards were stored at −80 °C until use.

#### 2.7.3. YFV RT–qPCR

The quantification of the YFV genome from the total RNA extracted from the mosquito pools was performed using RT–qPCR [43] together with the standard curve prepared in the previous step. The reaction was performed using the Superscript III^®^ Platinum^®^ One-Step Quantitative RT–PCR System kit (Invitrogen) in a total volume of 25 μL, containing 12.5 nM of each primer (YfallF-GCT AAT TGA GGT GYA TTG GTC TGC and YfallR-CTG CTA ATC GCT CAA MGA ACG CAC), 5 nM of probe (YfallP-ATC GAG TTG CTA GGC AAT AAA, labeled with FAM and BHQ1) and 5 μL of the extracted total mosquito RNA. The runs were performed in a 7500 real-time PCR system (Applied Biosystems, Waltham, MA, USA). The cycling conditions were as follows: 50 °C for 30 min, followed by a 2-min cycle at 95 °C, then 45 cycles of 15 s at 95 °C and 1 min at 60 °C. Samples with CT values below 37 were considered positive. The YFV genome concentration in the samples was determined in RNA copies/µL using a standard curve in the Applied Biosystems 7500 Software version 2.3. The values were then converted into copies/mg of mosquito using the following formula:$$Q=\frac{\left(\frac{q\times 60\times 1000}{140}\right)}{10\times 1}$$
where,

*Q* = quantification in copies/mg,

*q* = quantification in copies/µL,

60 = RNA elution volume,

1000 = volume of PBS used in the maceration,

140 = volume of macerate used in the extraction,

10 = number of mosquitoes per pool,

1 = mean mass of 1 mosquito in mg.

### 2.8. Ethical Aspects

This study was approved by the directors of the CENP (concession of animals), the Committee on Ethics in the Use of Animals of the IEC (CEUA/IEC—protocol no. 0030/2019) and the Biodiversity Authorization and Information System of Instituto Chico Mendes de Biodiversity Conservation (SISBIO/ICMBio—protocol no. 61757-1).

## 3. Results

*Ae. albopictus* were infected with YFV after feeding on *Callithrix* experimentally infected with YFV. Engorged female *Ae. albopictus* were analyzed at 14 and 21 dpi. The presence of viral replication was detected by viral isolation in C6/36 cells from the mosquito saliva, head, thorax/abdomen and leg samples, and these results were confirmed by simultaneous molecular detection. Thus, the laboratory transmission of YFV by these mosquitoes to PNHs of the genus *Callithrix* was confirmed, as well as the infection of *Ae. albopictus* by YFV after the blood meal in previously infected NHPs.

All arthropods were subjected to viral isolation. The first-passage C6/36 cells were used for the detection of infection in *Ae. albopictus* saliva samples. Saliva samples from 60 mosquitoes were analyzed and organized into batches of 10 specimens, with 3 batches from 14 dpi (30 mosquitoes) and 3 from 21 dpi (30 mosquitoes). Of the saliva samples from 14 dpi, 66.7% (2/3) were positive for the presence of YFV. For the saliva samples from 21 dpi, 33.3% (1/3) were positive for YFV (Figure 2).

Infection is a prerequisite for the spread of the virus, and in the present study, the presence of YFV was also analyzed in batches of the head, thorax/abdomen and legs, with 10 specimens in each sample. A total of 21 lots were analyzed, 9 at 14 dpi and 12 at 21 dpi. Three negative control samples (referring to the same body parts tested) were tested for each analyzed group (Table 1). One hundred percent of the samples from the head, thorax/abdomen and legs were positive for YFV at 14 dpi. For the samples, 58% (7/12) were positive for YFV at 21 dpi (Figure 3).

For the molecular analysis, 70 female *Ae. albopictus* were used to make 21 head, thorax/abdomen and leg batch samples, with 9 samples (30 specimens) at 14 dpi and 12 samples (40 specimens) at 21 dpi. The RT–qPCR results indicated that all samples at 14 dpi were positive, with the CT values ranging from 23.5 to 17.6, and that for the samples at 21 dpi, 58% (7/12) were positive and presented CT values between 34.2 and 15.4. An absolute quantification of the YFV genome was also performed for all samples (Table 1).

Blood samples from the 10 NHPs were analyzed by RT–qPCR. NHP1, NHP2 and NHP3 (infected by inoculation) were positive for YFV; NHP4, NHP5 and NHP6 were negative (blood meal on 14 dpi); and of the animals exposed to *Ae. albopictus* at 21 dpi (i.e., NHP7, NHP8 and NHP9), only 1 animal (NHP9) was positive, with a CT of 8.4 (Table 2).

Correlating the results of the analyses of the NPHs and mosquitoes used in the study, it was possible to observe that all the Culicidae samples at 14 dpi were positive both by viral isolation and by RT–qPCR, while the primates (NHP4, NHP5 and NHP6) that were fed by these mosquitoes at 14 dpi showed negative results by RT–qPCR. A total of 58% (7/12) of the Culicidae samples at 21 dpi were positive by RT–qPCR and viral isolation, and one of the primates fed during that period (NHP9) was positive for YFV by RT–qPCR.

## 4. Discussion

Evaluations of the vector competence of *Ae. albopictus* in the transmission of YFV to NHPs of the genus *Callithrix*, to the best of our knowledge, have not yet been performed, which makes the data obtained in the present study relevant, especially considering that risk assessments are necessary in the context of the identification of potential vectors for the virus and for the urbanization of YF in Brazil. Furthermore, the presence of YFV confirmed in one of the primates reinforces the relevance of the present study.

Urbanization increases the larval habitats of *Ae. albopictus* and accelerates the development and survival of mosquitoes. In the last 20 years, the area inhabited by this culicid has increased significantly, which, in turn, has potentially increased the ability of the vector to transmit diseases. The reurbanization of YF has been much discussed, especially in terms of the true susceptibility and ability of *Ae. albopictus* to become infected and act as a link vector between the sylvatic cycle and the urban environment [28,44,45,46].

Since the reemergence of YF in 2014/2015, Brazil has recorded its largest YF epidemic in recent decades, affecting mainly the Southeast region. Abreu et al. [47] demonstrated the risk of reurbanization and the seasonal reemergence of YFV in a study conducted in the state of Rio de Janeiro, where they found a female of the genus *Alouatta* infected with YFV and *Ae. albopictus* trying to bite the dead animal’s body, thus emphasizing the need for continuous and effective surveillance, in addition to high vaccination coverage.

The transmission of arboviruses from a vector to a vertebrate host usually occurs through the saliva expelled by the mosquito during blood feeding [48]. In the present study, YFV was found in the saliva of *Ae. albopictus* in 100% of the samples analyzed at 14 dpi and 21 dpi. In studies conducted by Amraoui et al. [2], YFV was detected in the saliva of *Ae. albopictus* after 4 passages in cells; the authors suggest that the culicid represents a threat in the context of the reurbanization of YF.

The presence of YFV in the saliva of *Ae. albopictus* indicates the potential for this Culicidae, adapted to the periurban environment, to transmit YF in urban areas in the countries where it is present. Additionally, there should be concern about the transmission of other arboviruses by this vector, such as those that cause dengue, Zika and Chikungunya, in the urban environment [19].

By viral isolation and molecular detection, YFV was found in different parts of the body of *Ae. albopictus,* both at 14 and 21 dpi, indicating the dispersion and infection of the mosquitoes, in addition to suggesting 14 dpi as the best period for understanding the kinetics of the vector infection based on the results presented in this study. It is relevant to highlight that the legs of Culicidae are important for understanding the dissemination of the virus in arthropod organisms. Our results indicated that the virus replicates in all parts of the mosquito’s body, including the legs.

Although this Culicidae has not been indicated as a YF vector in Brazil, in a study carried out by researchers from the Evandro Chagas Institute (IEC/PA) in 2017, YFV was found in samples of *Ae. albopictus* from the southeastern region of the country; however, more studies on the natural infection need to be developed. Experimentally, this vector has already been infected with YFV [29,49,50].

It is necessary to combat this mosquito through a better understanding of its biology to allow for the implementation of new effective control strategies. A metagenomic analysis of *Ae. albopictus*, for example, may allow for comparative analyses and even the discovery of new genes and elements that may be useful for innovative genetic control strategies for this mosquito, which is a potential vector of important viruses that can affect public health [51,52].

The genus *Callithrix* was used as an experimental model due to its periurban and urban habits as well as ease of handling. In this study, one of the primates (NHP9) was positive, detected by molecular analysis, for YFV, a finding that suggests the transmission of YFV by mosquitoes via blood feeding at 21 dpi. Although YFV was detected both by viral isolation and by molecular analysis in all samples at 14 dpi, only at 21 dpi was there transmission of YFV to NHP9. However, it is possible to suggest that the age of mosquitoes at 14 dpi is related to the virus transmission processes, although it is relevant to note that a greater number of mosquitoes, as well as further studies, would be necessary to confirm this hypothesis. Furthermore, at the 14th dpi, there was no transmission to the NHP, as occurred at the 21st dpi, suggesting that, in addition to the viral load, other factors may have interfered in this process.

The transmission of arboviruses by mosquitoes depends on the genetic characteristics of the vectors and the virus, as well as on several factors such as the incubation temperature, viral load, and stress conditions. More studies and investigations need to be performed to elucidate the roles of these factors.

The detection of epizootic diseases in NHPs by surveillance services is essential for surveillance activities and for the diagnosis and detection of cases in humans, enabling better epidemic control [46].

In recent years, entomovirological surveillance has proven to be a useful tool for the detection of medically important viruses in the event of outbreaks, the monitoring of virus circulation and for characterizing vectors, which are fundamental elements for understanding the dynamics of vector-borne diseases. Vector competence studies in different mosquito species are important. The data on vectors and viruses associated with human diagnoses are essential for establishing disease control strategies [2,7,16].

The risk of new YF outbreaks is high in the region of the Americas, especially considering the impacts of the COVID-19 pandemic, which led to a decrease in the proportion of the population vaccinated against YF; at national levels, YF vaccination coverage was 57% in 2021 [3].

Entomological data, continuous epidemiological surveillance and the analyses of landscape composition can contribute to the protection of susceptible areas to possible outbreaks of YF, allowing for protective measures to be taken to avoid cases in humans [53].

There is a need for continuous effective surveillance and for an increase in vaccination coverage, as vaccination is the most effective preventive measure available for YF. Entomological surveillance actions should be considered for the control of *Ae. albopictus*, especially in urban areas. Control efforts should therefore be strengthened and maintained to prevent the urbanization of YF and new outbreaks of the disease.

## 5. Conclusions

The detection of YFV in the saliva, head, thorax/abdomen and legs confirms that *Aedes albopictus* can become infected through blood feeding on NHPs, suggesting the competence of this Culicidae as a potential vector for YFV. The positive results for the presence of YFV in NHPs of the genus *Callithrix* after exposure to *Aedes albopictus* indicates that this mosquito serves as a transmission vector of YFV, reinforcing the relevance of the present study and the need for conducting research, especially with the goal of conducting natural and continuous entomological surveillance.

## Figures and Tables

**Figure 1 viruses-15-01019-f001:**
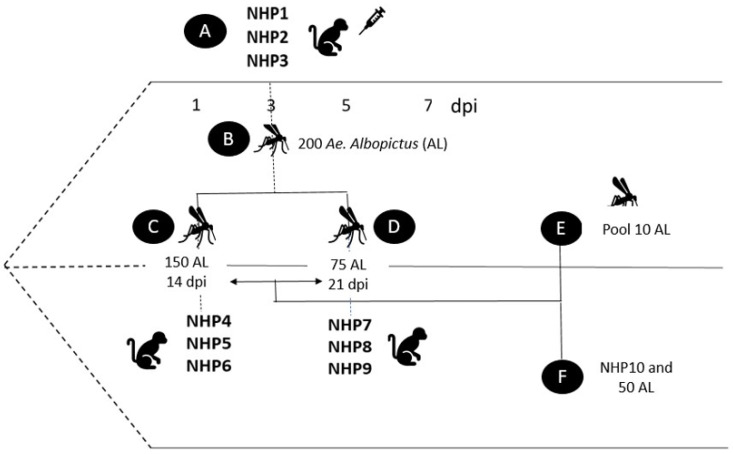
Experimental procedure. Legend: (**A**) Experimental infection of non-human primates (NHP1, NHP2 and NHP3) with YFV; (**B**) NHP exposure to female *Ae. albopictus* (AL) on 3 dpi; (**C**) Exposure of healthy NHPs (NHP4, NHP5 and NHP6) to female *Ae. albopictus* on 14 dpi and segmentation of females into thorax, abdomen and head and extraction of saliva; (**D**) Exposure of healthy NHPs (NHP7, NHP8 and NHP9) to females *Ae. albopictus* on 21 dpi and segmentation of females into thorax, abdomen and head and extraction of saliva; (**E**) Segmentation of female (in thorax, abdomen, head) and saliva extraction, all samples organized in pools of 10 at 14 and 21 dpi; (**F**) NHP (negative control) and 50 mosquitoes used as negative controls.

**Figure 2 viruses-15-01019-f002:**
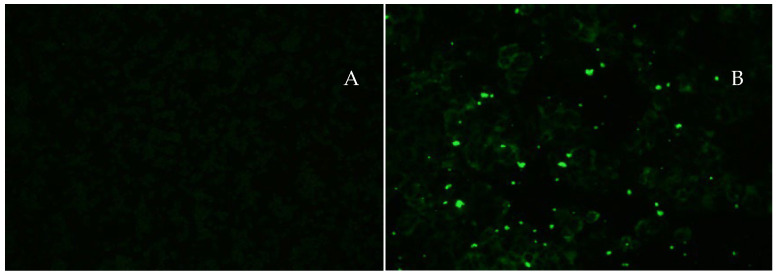
Indirect immunofluorescence using polyclonal antibodies specific for YFV in C6/36 cell cultures: (**A**) negative control; (**B**) positive saliva pool sample (1AALS) on 14 dpi. The magnification is 200×.

**Figure 3 viruses-15-01019-f003:**
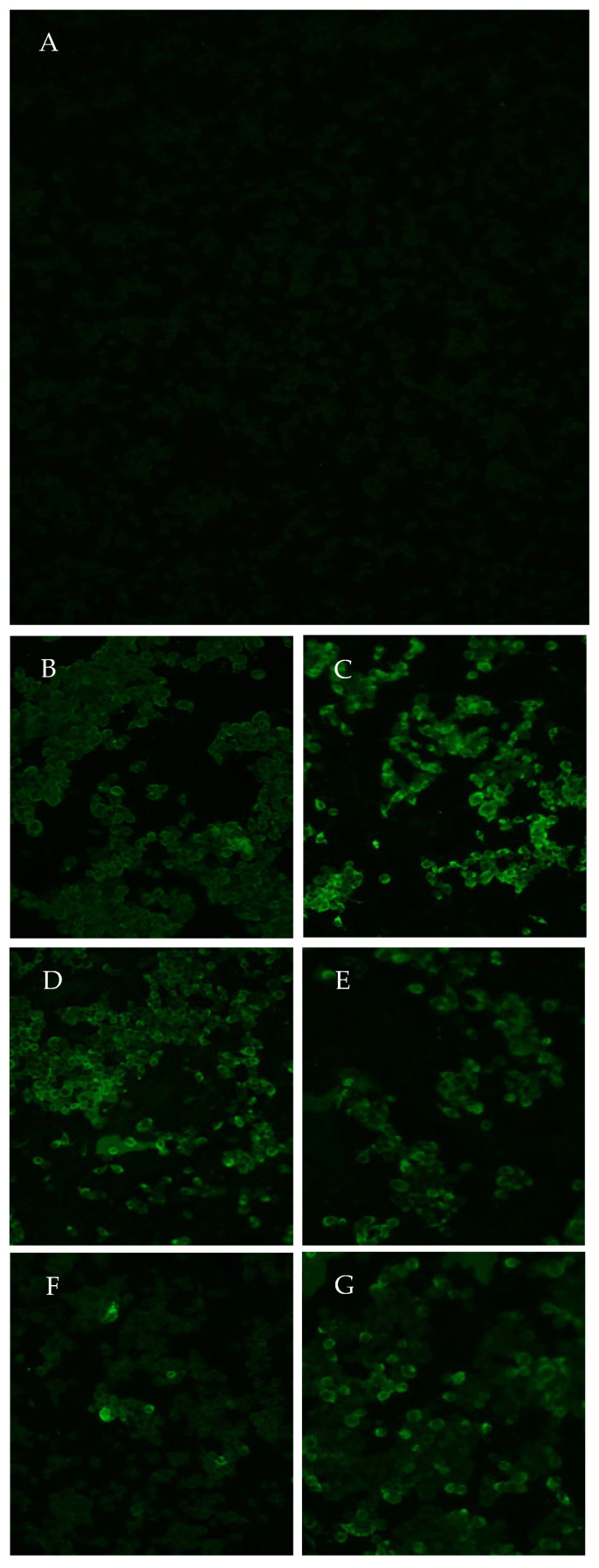
Indirect immunofluorescence using monoclonal antibodies specific for YFV in C6/36 cell cultures: (**A**) negative control; (**B**) positive head sample (2AALC) analyzed at 14 dpi; (**C**) positive head batch sample (3AALC) analyzed at 21 dpi; (**D**) positive thorax/abdomen batch sample (2AALT) analyzed at 14 dpi; (**E**) positive thorax/abdomen batch sample (3AALT) analyzed at 21 dpi; (**F**) positive leg batch sample (2ALP) analyzed at 14 dpi; (**G**) positive leg batch sample (3AALP) analyzed at 21 dpi. The magnification is 40×.

**Table 1 viruses-15-01019-t001:** *Ae. albopictus* batches, dpi, cell culture results, and RT–qPCR quantification of YFV.

Ae. albopictus (Batch)	DPI	Isolation in Cell Culture	RT–qPCR	CT	Quantification (Copies/mg)
1AALT	14	Positive	Positive	18.2	>1.43 × 10^6^
1AALC	14	Positive	Positive	20.8	2.92 × 10^5^
1AALP	14	Positive	Positive	23.5	4.81 × 10^4^
2AALT	14	Positive	Positive	18.7	1.19 × 10^6^
2AALC	14	Positive	Positive	22.0	1.30 × 10^5^
2AALP	14	Positive	Positive	23.8	3.95 × 10^4^
3AALT	14	Positive	Positive	17.6	>1.43 × 10^6^
3AALC	14	Positive	Positive	20.8	2.85 × 10^5^
3AALP	14	Positive	Positive	20.4	3.82 × 10^5^
1AALT	21	Positive	Positive	20.3	4.05 × 10^5^
1AALC	21	Positive	Positive	18.1	1.75 × 10^6^
1AALP	21	Positive	Positive	20.4	3.74 × 10^5^
2AALT	21	Negative	Negative	>37	0
2AALC	21	Positive	Negative	>37	0
2AALP	21	Negative	Negative	>37	0
3AALT	21	Positive	Positive	15.4	>1.43 × 10^6^
3AALC	21	Positive	Positive	16.4	>1.43 × 10^6^
3AALP	21	Positive	Positive	19.4	7.12 × 10^5^
4AALT	21	Negative	Negative	>37	0
4AALC	21	Negative	Positive	34.2	<1.43 × 10^2^
4AALP	21	Negative	Negative	>37	0

Legend: DPI = days post-infection; AALT = batch sample, thorax/abdomen; AALC = batch sample, head; AALP = batch sample, legs; CT = cycle threshold.

**Table 2 viruses-15-01019-t002:** Non-human primates of the genus *Callithrix* exposed to *Ae. albopictus* infected with YFV.

Primate	DPI	RT–qPCR	CT	Quantification
NHP1	INOC	Positive	28.1	463.4
NHP2	INOC	Positive	22.5	17,882.1
NHP3	INOC	Positive	26.2	671.8
NHP4	14	Negative	NA	NA
NHP5	14	Negative	NA	NA
NHP6	14	Negative	NA	NA
NHP7	21	Negative	NA	NA
NHP8	21	Negative	NA	NA
NHP9	21	Positive	8.4	320,019,296.0
NHP10	NC	Negative	NA	NA

Legend: NHP = non-human primate; DPI = days post-infection; INOC = inoculation; CT = cycle threshold; NC = negative control; NA = not applicable.

## Data Availability

Not applicable.

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
