# Peer review of "Vector Competence of Aedes albopictus for Yellow Fever Virus: Risk of Reemergence of Urban Yellow Fever in Brazil"

_viruses, 2023, doi:10.3390/v15041019_

Round 1

Reviewer 1 Report

Review of Caldeira et al, Viruses 2023, 15, x. https://doi.org/10.3390/xxxxx

This paper by Caldeira et al tests the vector competence of low generation (F5) Aedes albopictus mosquitoes from Brazil for YFV by infecting them via direct blood feeding on infected marmosets that were inoculated with virus from cell culture and demonstrating horizontal transmission via blood feeding to 1 of 6 exposed marmosets. The work is important because it adds to the evidence base that this mosquito species could vector YFV, following on laboratory-based vector competence studies using the same vector and virus pairing, some field evidence of naturally-infected A. albopictus that seems to be from the authors (but this doesn’t seem to be peer-reviewed?…Reference 50…it is not clear what this reference is to so the authors should show address this), and prior publications from Brazil that urban/peri-urban marmosets have become infected or seroconverted to arboviruses that are spread by their human-associated mosquitoes. The study was conducted fairly-well, with decent replication and use of negative controls, and the authors should be commended for doing a transmission study from NHP-to-mosquitoes-back-with an animal that may serve as both an animal model and a true reservoir in nature, and using low generation local mosquitoes and a local virus genotype. The paper is mostly well-written, but it is unclear in some parts and the study overall does have with flaws that should be addressed (such as missing statistical analyses and instead focusing on qualitative results, lack of positive controls, lack of clinical data on the NHPs, and not acknowledging and pointing out limitations of the study in the Discussion). Overall, the evidence is good that A. albopictus can transmit YFV and that marmosets can get infected by their infectious bites, but it fails to address the efficiency of this vector competence mostly because of the study design (e.g. testing only pools and other issues).  Specific edits that should be addressed are listed below:

Line 23: change to “diseases”

Line 27 and in the Methods: I don’t see where the authors identify the species of Callithrix in the paper.  This should be done. They say that the virus was isolated from C. penicillata, but don’t say that this was the species used in the experiments.

Line 28: Change to “Female Ae. albopictus were first blood fed on Callithrix ____, a non-human primates (NHP) that was infected with YFV via needle inoculation.”

Line 33 and elsewhere: Genus species…species should not be capitalized

Line 64: It is not correct that A. albopictus has not been implicated as an arbovirus vector in Brazil.

Line 123 & 179: Change to, “…A. albopictus cells (clone C6/36)”

Line 299: “samples”

Line 328: Please rewrite the following to be more clear on what you mean: “….Ae. albopictus in the environment by probing the animal's body”

Line 344 & 345: These sentences do not make sense and are wrong.  This limited experiment does not suggest 14dpi is the best period for understand the kinetics of YFV, and mosquito legs are not the fundamental to the spread of viruses in arthropods. 

Lines 353-8: This paragraph doesn’t make sense relative to the structure of the paper, nor does it add anything.  I would suggest it be deleted.

Line 365: This is incorrect.  One cannot say there was a lower viral load in the mosquitoes at 14dpi as evidence that they may have transmitted more inefficiently to the marmosets. Arboviral load in mosquito body tissues does not correlate well or at all with titers expectorated in the saliva during biting, and the experiment blood fed many mosquitoes on the NHPs but did not test the mosquitoes’ saliva titer after biting.

Overall: There should be a paragraph in the Discussion that discusses the limitations of the study. Below are the limitation that I see that should be discussed and/or fixed:

·        The analysis of mosquito tissues was always grouped into pools which prevented the authors from examining the efficiency of this mosquito’s vector competence for YFV

o   This also prevented any statistical analysis of the vector competence assays.

·        Positive controls in the cell culture experiments (infecting the cells directly with virus stocks or with mosquitoes that were IT inoculated and showing pictures of these results) were not done or presented. These positive control data would lend more confidence in the results.

·        Not serially sampling the blood of the NHPs often may have led the authors to miss a transient viremia

·        Biosafety levels were not discussed

·        There is no presentation of clinical assessments or clinical data from the marmosets from monitoring which would have provided valuable information and also followed ethical guidelines on NHP use in the study.  For example, what was the titers in the NHPs post-inoculation and what were their clinical signs?, and how was their disease handled?, were they euthanized or did the infections resolve? Also, NHP9 had a very high viremia 3.2x108 genomes, which suggest severe disease.  

Reviewer 2 Report

The authors report a study of experimental infection of the yellow fever virus in Ae. albopictus from previously inoculated Callithrix sp.. In addition, they also report the infection of a Callithrix after the bite of infected Ae. albopictus. In short, they reinforce the importance of Ae. albopictus as a potential YFV vector in Brazil.  A more accurate picture of the methodology could be inserted to facilitate the understanding of the experimental protocol (especially number and type of samples tested in each phase). Other comments can be found bellow.

Line 33: Please, correct to “Ae. albopictus”

Line 125: Is it possible to determine “Viral load” using RT-qPCR? Using a standard dilution curve? Please, clarify.

Lines 130 to 136: NHPs were "shaved"? Did the mosquitoes feed well?

Line 169 and Figure 1: It was not clear to me how many females were "salivated" or when this was done. It is mentioned that it occurred on days 14 and 21pi, but the figure shows salivation only after 21dpi. Was salivation performed after females fed in the NHPs?

Line 264: Is it possible to know the viral load in NHPs at 3dpi?

Line 269: Please, italicize “Callithrix”.

Line 268-269: Do the cited results not demonstrate the opposite? The infection of Ae. albpictus from infected Callithrix?

Lines 271-275: It could be better described during the method section.

Lines 274-275: Why does saliva infection seem to decrease after 21dpi, compared to 14dpi?

Line 286: Again, it is important to discuss why infection rate goes down after 14dpi.

Line 306: Please, correct “Ae. albopictus”

Line 325: Maybe the reference [48] should be this one: 10.1590/0074-02760190076, which shows Ae. albopictus on Alouatta.

Line 348 to 350: Are these results published? I didn't find it in references 50 and 51.

Line 359 to 367: Why do the authors think so few Callithrix were infected? Was the inoculum of NHP1 to 3 much higher than the viral load found in mosquito saliva?

Reviewer 3 Report

This work evaluates the vector competence of Ae. albopictus collected in Ananindeua/PA when fed on NHP infected with YFV isolate from Callithrix penicillata. Samples from the body and thorax, legs and saliva of mosquitoes were analyzed. There was a possibility of infection, dissemination and transmission of YFV by this species of Culicidae.

______________________________________________________________________

Titer

Line 1-2. Suggestion, consider inserting the NHP in the title, as some works have already shown that Ae. albopictus is a possible vector for YFV.

Resume

Line 22. Standardize Yellow fever virus (YFV) nomenclature.

Line 24. Standardize Yellow fever virus (YFV) nomenclature.

Line 27 – 28. Suggestion would be to change the sentence because the way it was written it seems that the NHP were infected by oral feeding.

Line 33 - 34. Standardize Yellow fever virus (YFV) nomenclature.

Introduction

Line 38. Standardize Yellow fever virus (YFV) nomenclature.

Line 41. Standardize Yellow fever virus (YFV) nomenclature.

Line 44. Standardize Yellow fever virus (YFV) nomenclature.

Line 54.  Failed to insert reference.

Line 77.  Standardize Yellow fever virus (YFV) nomenclature.

Methodology

line 120 - 125. It is not mentioned how many passages were necessary to reach the title used in the experiment.

Line 126. I don't know if you misconfigured the image but it was a little confusing to understand some details of the scheme. The letter A was on top of the number 142 (I didn't understand what it does there, and the other numbers -143 and 150). In B the mosquito image has been cropped. In the legend the letter E appears before the D.

Why did you feed the mosquitoes at 14 and 21 dpi with NHP? Mosquitoes would survive on glucose alone up to 21 dpi. This work was not intended to assess vertical transmission (which would justify blood feeding). In the work, it is not clear that the objective of blood feeding at 14 and 21 dpi could be another way of evaluating transmission via saliva.

Line 70. Why use head and legs to assess dissemination? Detection in heads is more accurate as some of the six legs can be lost at the time of dissection and can influence viral detection. Is it true that for all mosquitoes there were six legs?

Line 278. Figure 2. The letters are not in the correct order (letter A does not appear).  The image corresponding to the letter C cannot clearly visualize the infected cells.

Line 272- 273. Why didn't you do the viral detection in saliva individually? There are studies that managed to detect YFV in the saliva of a single mosquito. In this way, the number of infected samples is displayed more accurately.

Line 280. Is the image really at 400x magnification?

Line 291. Figure 3. It looks like the F word is smaller in size than the other words.

Results

Suggestion is to insert correlation between the viral load of the NHP and the load of the head, legs, and abdomen samples. Could it be that in this case the viral load influenced the positivity of the different samples?

Discussion

Line 324. Standardize Yellow fever virus (YFV) nomenclature.

Lines 362-363. It would take a larger number of mosquitoes to assume that the age of mosquitoes at 14 dpi is or is not related to transmission in the host (which was not the case in this research).

Line 381. Standardize Yellow fever virus (YFV) nomenclature.

Line 383. Standardize Yellow fever virus (YFV) nomenclature.

Line 387. Standardize Yellow fever virus (YFV) nomenclature.

Line 389. Standardize Yellow fever virus (YFV) nomenclature.

Line 392. Standardize Yellow fever virus (YFV) nomenclature.
